# Curiosity or savouring? Information seeking is modulated by both uncertainty and valence

**Lieke L. F. van Lieshout**[1,2]*, **Iris J. Traast**[1], **Floris P. de Lange**[1], **Roshan Cools**[1,2]

**1** Donders Institute for Brain, Cognition and Behaviour, Radboud University, Nijmegen, The Netherlands,
**2** Department of Psychiatry, Radboud University Medical Centre, Nijmegen, The Netherlands

* l.vanlieshout@donders.ru.nl

## Abstract

Curiosity is pervasive in our everyday lives, but we know little about the factors that contribute to this drive. In the current study, we assessed whether curiosity about uncertain outcomes is modulated by the valence of the information, i.e. whether the information is good or bad news. Using a lottery task in which outcome uncertainty, expected value and outcome valence (gain versus loss) were manipulated independently, we found that curiosity is overall higher for gains compared with losses and that curiosity increased with increasing outcome uncertainty for both gains and losses. These effects of uncertainty and valence did not interact, indicating that the motivation to reduce uncertainty and the motivation to maximize positive information represent separate, independent drives.

## Introduction

Curiosity is a strong driver of behaviour, but relatively little is known about its psychological and neurobiological mechanisms [1]. It is often defined as a motivational state that stimulates exploration and information seeking to reduce uncertainty [1–4]. Obtaining this information can be useful to increase rewards or to make better decisions, for example when we check the stock market to see if we should make a new investment. When seeking such directly relevant information, we hope to maximize reward or minimize harm (i.e. not wasting your money on useless investments; see [5]). The type of information search that is aimed at reward maximization is commonly studied in literature focusing on the explore-exploit tradeoff or goal-directed exploration [6–11] and can be referred to as "instrumental curiosity".

In contrast to maximizing rewards, humans and other animals are also curious about information that does not serve a direct, obvious benefit [12–16]. Think for example about situations in which we scroll through our Instagram feed without a specific purpose. This type of curiosity is referred to as "non-instrumental curiosity" (see also [1]).

Why would we be curious about such purposeless information? According to the information gap theory [17], curiosity generally arises when we become aware of gaps in our knowledge and we are driven to fill these gaps with information. Indeed, recent studies have indicated that we are mainly curious about information, because it might reduce uncertainty about the world around us [13, 14, 16, 18, 19]. For instance, in our recent study, we used a

Science Framework (https://osf.io/h8r6f/
registrations).

**Funding:** This work was supported by The
Netherlands Organisation for Scientific Research
(NWO Vidi award 452-13-016 to FPdL and NWO
Vici award 453-14-015 to RC), the James
McDonnell Foundation (JSMF scholar award
220020328 to RC) and the EC Horizon 2020
Program (ERC starting grant 678286 awarded to
FPdL). The funders had no role in study design,
data collection and analysis, decision to publish, or
preparation of the manuscript.

**Competing interests:** The authors have declared
that no competing interests exist.

lottery task in which we independently manipulated the uncertainty about the outcome of a lottery, as well as the expected value of that lottery. Participants were asked to indicate their curiosity about the lottery outcome. They were instructed that the information provided by the outcome was non-instrumental. That means that all outcomes were obtained regardless of their curiosity decisions and there was no way to maximize their rewards during the task. The results indicated that when the uncertainty about lottery outcome was higher, participants exhibited higher curiosity ratings and greater willingness to wait for information about those outcomes [16]. These findings suggest that the drive for information is a function of the size of an information gap [17] rather than merely the likelihood of reward maximization. This concurs with the proposal that curiosity represents a drive to improve our knowledge about what is going to happen [16].

Recent evidence indicates that the drive to seek information might be asymmetric, with people exhibiting a preference for positive over negative belief updating, such that humans are more curious about positive than negative information [12, 20]. In fact, human volunteers have been shown to be willing to pay for ignorance about negative information as well as for knowledge about positive information [12]. This is consistent with the observation that people often prefer not to be informed about potential negative medical test results [21, 22]. It should be noted that this preference for positive over negative belief updating was found in situations when participants had no influence on their payoffs. This is in contrast with findings from the explore-exploit literature showing that when participants can influence their payoffs, they show higher exploration rates for losses compared with gains, reflecting intensified alertness in the face of potential losses [10, 11].

These observations led us to ask whether people's drive for information in a non-instrumental context depends on the valence of the information gap, with greater curiosity for greater positive information gaps, but reduced curiosity for greater negative information gaps. This preference of reducing positive uncertainty, but accepting negative uncertainty bears resemblance to the well-known framing effect in prospect theory [23, 24], where people are risk averse for moderately probable gains, but risk seeking for moderately probable losses. Specifically, we aimed to investigate whether the uncertainty of the lottery outcome had contrasting effects on curiosity about information depending on its valence.

## The present experiments

To investigate this, we designed a set of experiments by adapting a lottery task we used previously [16]. In this task, participants were presented with a lottery consisting of a vase containing a mix of red and blue marbles, which are associated with different monetary values. Participants were instructed that one marble would be randomly selected from the vase, and that they would gain/lose the money associated with the marble. This lottery task enabled independent manipulation of the uncertainty and expected value of lottery outcomes, as well as lottery outcome valence (whether the lottery contained gains or losses). In other words: each lottery was associated with more or less uncertain gains and losses.

In Experiment 1A, 1B and 1C, participants were asked to indicate how curious they were about the outcome of a presented lottery in terms of self-report ratings. Importantly, participants could not influence the actual outcome and whether they would see the outcome or not. Experiment 2 was conducted to verify the robustness of these explicit curiosity ratings by investigating curiosity more implicitly in terms of willingness to wait decisions. The design of Experiment 2 was similar to that of Experiment 1A, 1B and 1C, but instead of explicitly rating their curiosity, participants had to indicate whether they wanted to see the lottery outcome or not. In this task, participants could control whether they would see the outcome, but this came

with a cost: participants had to wait an additional 3–6 seconds if they wanted to see the outcome. We used willingness to wait because it is a well-established measure of an item's motivational value [25], and it has previously been linked to curiosity [16, 20, 26]. This underlines the notion that waiting is costly, in line, for example, with the intertemporal choice literature. In both experiments, participants were clearly instructed that the information provided by the outcome was non-instrumental: all outcomes were obtained regardless of participants' curiosity or willingness to wait decisions and they could not influence how much money they would gain or lose during the task.

Two hypotheses were considered. The null hypothesis is that people's drive for information is independent of the valence of that information. If this is true, the effect of outcome uncertainty on curiosity should not be a function of gain versus loss outcome valence. The alternative hypothesis is that people's drive to update their beliefs is positively biased, with a preference for updating positive versus negative beliefs. If this is the case, then any effect of outcome uncertainty should vary with the valence of the lottery outcome. According to the strong version of this alternative hypothesis, participants would exhibit greater curiosity for more uncertain positive outcomes, but reduced curiosity for negative outcomes with greater uncertainty. Previous studies have addressed the existence of multiple motives for curiosity. Different individuals exhibit various mixtures of effects of uncertainty and expected value, both for gains as well as for losses [15]. Additionally, previous work has demonstrated a boost in information seeking when outcome probability was most uncertain [12], both for gains as well as for losses. However, these studies have not explicitly addressed the interaction between uncertainty and valence on non-instrumental curiosity. It is this interaction that allows us to investigate the degree to which the effects of uncertainty and valence represent independent or interactive mechanisms of non-instrumental curiosity.

To preview, the results of the curiosity experiments (Experiment 1A, 1B and 1C) demonstrated that while curiosity was overall higher for gains, curiosity increased with outcome uncertainty for both gains and losses. Valence and uncertainty affected curiosity largely independent from each other, given that the effect of outcome uncertainty was not reliably different between gains and losses. These results were replicated with the more implicit willingness to wait measure (Experiment 2), indicating that people were willing to pay with time to satisfy their curiosity. On average, participants were willing to wait in 45.4% of all trials. These results suggest that curiosity is elicited by uncertainty about what will happen and that people are driven to improve this model of the task, regardless of outcome valence. In addition, there is a bias towards information gain about positive compared with negative information, which operates independently of uncertainty.

## Methods

We conducted four experiments using lottery tasks. In Experiment 1A, we investigated participants' curiosity about the lottery outcome explicitly using curiosity ratings. We performed two additional experiments, Experiment 1B and Experiment 1C to investigate whether the results of Experiment 1A are replicated if the gain and loss trials are not presented in separate blocks, but vary on a trial-by-trial basis instead. By manipulating gain and loss in a randomized design, we control for nonspecific state-related changes (i.e. mood or arousal) associated with the gain or loss context. Participants of both experiments performed the same task, with the only difference that the participants of Experiment 1B performed the experiment in a behavioural lab, whereas participants of Experiment 1C did the experiment while we non-invasively recorded neural activity using fMRI. The neural results are outside the scope of the current manuscript, so we will not discuss details regarding fMRI procedures.

Experiment 2 was conducted to verify the robustness of these explicit curiosity ratings by investigating curiosity more implicitly in terms of willingness to wait decisions. The information provided by the outcome was non-instrumental in both tasks: all outcomes were obtained regardless of participants' decisions and they had no way of influencing how much they would gain or lose during the task.

## Data & code availability and preregistration

All data and code used for stimulus presentation and analysis are freely available on the Donders Repository (https://doi.org/10.34973/5bzr-kb30). Experiment 1A and its analyses were preregistered on the Open Science Framework (https://osf.io/h8r6f/registrations).

## Participants

**Experiment 1A.** Thirty-seven healthy individuals participated in Experiment 1A, that involved curiosity ratings and gain/loss blocks. Two participants were excluded because they reported during the debriefing that they (falsely) believed that they could influence the lottery outcome. It should be noted that this crucial aspect of the task was stressed multiple times during instructions, so these participants evidently failed to understand the task. We therefore excluded them based on this preregistered criterion. One participant was excluded due to a lack of variation in responses (this participant gave the same curiosity rating on every trial). Though the lack of variation in responses was not preregistered, we decided to exclude the participant because no models could be fitted on the participant's data and because we could not ensure that the participant was engaged with the task. The final sample of Experiment 1A consisted of thirty-four participants (27 women, age 21.79 ± 4.13, mean ± SD). We were aiming for a sample size of N = 34 included datasets to detect a within-subject effect of at least medium size (d>0.5) with 80% power using a two-tailed one-sample or paired t-test.

**Experiment 1B.** Thirty-six healthy individuals participated in Experiment 1B, involving curiosity ratings and randomized gain/loss trials. Three participants were excluded. One participant was excluded because he reported during debriefing that he (falsely) believed that he could influence the lottery outcomes and the other participant because he did not give a curiosity rating in > 10% of all trials. The final sample of the experiment consisted of thirty-three participants (25 women, age 23.42 ± 3.40, mean ± SD).

**Experiment 1C.** Thirty-six healthy participants took part in Experiment 1C, involving curiosity ratings and randomized gain/loss trials while non-invasively recording neural activity using fMRI. Three participants were excluded. One participant was excluded due to excessive movement, one participant was excluded due to brain-peculiarities and another participant due to registration difficulties. The final sample thus consisted of thirty-three participants (25 women, age 23.14 ± 3.04, mean ± SD). All participants had normal or corrected-to-normal vision and had been screened for MR compatibility before the experiment started.

**Experiment 2.** Thirty-seven healthy individuals participated in Experiment 2, involving willingness to wait decisions and randomized gain/loss trials. We used the same exclusion-criteria as for Experiment 1A. Two participants were excluded because they reported afterwards that they (falsely) believed that they could influence the lottery outcome. Another participant was excluded due to a lack of variation in responses (gave the same response in > 98% of all trials). The final sample of the experiment consisted of thirty-four participants (22 women, age 22.59 ± 3.76, mean ± SD).

Participants of both experiments gave written informed consent according to the declaration of Helsinki prior to participation. The experiments were approved by the local ethics committee (CMO Arnhem-Nijmegen, The Netherlands) under a general ethics approval protocol

("Imaging Human Cognition", CMO 2014/288). The experiments were conducted in compliance with these guidelines. It should be noted that there was no overlap between participants across experiments. Participants who participated in one of the studies, were excluded from participation in any of the other experiments reported in the manuscript.

## Procedures

**Experiment 1A.** Each trial of the lottery task started with an image of a vase containing twenty marbles, each of which could be either red or blue (Fig 1A). In total, three vase configurations were possible: (1) 90%-10% vases: 18 marbles had one color and 2 marbles had the other color, (2) 75%-25% vases: 15 marbles had one color and 5 marbles the other color, (3) 50%-50% vases: 10 marbles had one color and 10 marbles the other color. Both colored marbles were associated with a monetary value that participants could gain or lose. Experiment 1A consisted of gain blocks, loss blocks and mixed blocks. In the gain blocks, both marbles were associated with positive monetary values; in the loss blocks, both marbles were associated

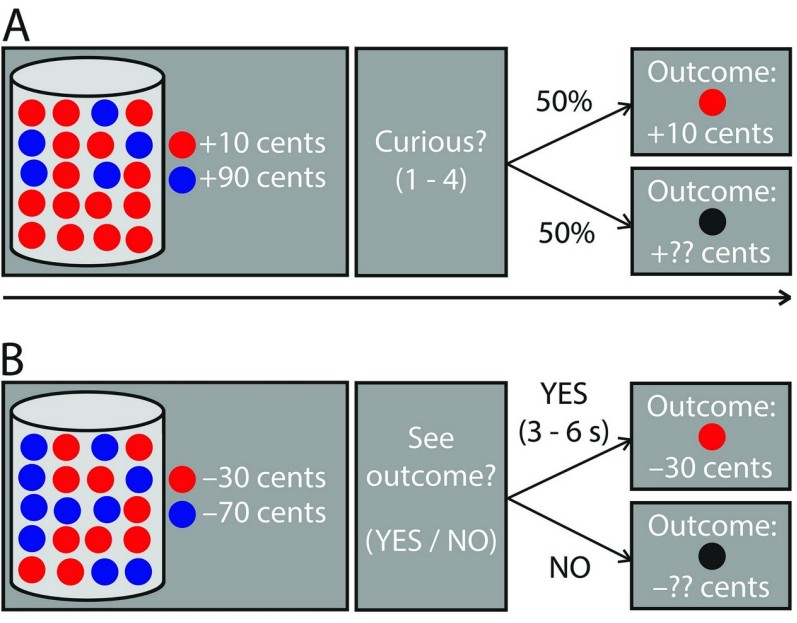

**Fig 1. Schematic depiction of the task of Experiment 1A, 1B and 1C (A) and Experiment 2 (B). A**. Schematic depiction of a gain trial in Experiment 1A, 1B and 1C. Participants saw a screen on which a vase with 20 marbles was depicted, either of which could be red or blue. Additionally, the monetary values associated with the red and blue marbles were depicted next to the vase. These monetary values could either both be positive (in gain trials, depicted here), both be negative (in the loss trials) or one value could be positive and the other one negative (in mixed trials, which were only present in Experiment 1A). Participants were told that one of the marbles would be randomly selected from the vase and that they would be awarded the money associated with this marble. Next, participants indicated how curious they were about seeing the outcome (1–4). There was a 50% chance of seeing the outcome on every trial, regardless of the participants' curiosity response. Importantly, a marble was randomly selected on every trial and participants were instructed that they were awarded the money associated with this marble, also if they would not see the outcome of a trial. See Methods (*2.3 Procedures*) for the timing the experiments. **B**. Schematic depiction of a loss trial in Experiment 2. Instead of giving a curiosity response, participants indicated whether they wanted to see the outcome of a trial or not. If they responded "No", the outcome was not presented to them. If they responded "Yes", the outcome would always be presented. However, they would then have to wait for an additional 3–6 seconds before they saw the outcome. Importantly, a marble was randomly selected on every trial and participants were instructed that they were awarded the money associated with this marble, also if they decided not to see the outcome of a trial. See Methods (*2.3 Procedures*) for details on the timing of Experiment 2.

with negative monetary values; and in the mixed blocks, one marble was associated with a positive monetary value and the other marble with a negative monetary value. Whereas the gain and loss blocks were of primary interest, we added the mixed blocks to investigate whether curiosity would be enhanced with added uncertainty about the valence of the outcome (i.e. whether people would actually gain or lose money). The monetary values varied on a trial-by-trial basis between +10 cents and +90 in gain blocks and between -90 and -10 cents in loss blocks (both in steps of 20 cents). All combinations of monetary values associated with the two differently colored marbles were possible, except for combinations of the same monetary values. In mixed blocks, the monetary value varied between -70 cents and +70 cents (in steps of 20 cents). One marble was always associated with a positive monetary value (gain) and the other marble with a negative monetary value (loss). The difference between these values was never higher than 80 cents. This was done to match outcome uncertainty (see Eq 1) between the three block types (gain/loss/mixed).

The participants were informed that on each trial, one marble would be randomly selected from the vase and that they would gain or lose the money associated with this marble. The first screen, on which the vase, the marbles and the monetary values associated with the marbles were depicted, was presented for 3 s (Fig 1A), followed by a blank screen (0.5 s). Next, a response screen was presented during which participants could indicate how curious they were about seeing the outcome of that trial ("How curious are you about the outcome?"). The curiosity scale ranged from 1–4. The response screen was presented until the participant responded, with a response limit of 4 s. Participants could give their responses on a button box using their right hand. They were instructed to use their index finger, middle finger, ring finger and little finger to indicate curiosity responses of 1, 2, 3 and 4 respectively. The response screen was followed by a blank screen (0.5 s) and an outcome screen (2 s). On each trial, participants had a 50% chance that their curiosity would be satisfied by seeing the outcome and a 50% chance that the outcome was withheld. This manipulation was instructed explicitly to participants and it uncoupled curiosity responses from the actual receipt of the outcome. This rendered an instrumental approach bias account of information seeking behaviour less likely ([27], see also [16]). The outcome screen depicted the vase, the marbles and monetary values associated with the marbles again, together with a box in which they saw the colored marble that was randomly selected and the amount of money they won or lost in that trial. When the outcome was not presented, participants saw a black marble and question marks at the location of the monetary value. After a trial ended, there was a blank screen with a jittered duration between 1 and 2 s (uniformly distributed).

Importantly, participants could not influence whether they would observe the outcome of a trial, or what the outcome of that trial would be. However, they were instructed that, on every trial, a marble would be randomly selected from the vase and that they would gain or lose the money associated with that marble, even if the outcome was not presented. They were told that the money they won or lost on every trial would be summed and that this sum of money would be added to or subtracted from the money they earned for their participation. This sum of money was depicted on the screen after the experiment had ended.

In total, the participants completed 3 blocks of every type (gain, loss and mixed), resulting in a total of 9 blocks. Each vase configuration was presented on 60 occasions, so 20 times for each block type. The participants completed a total of 180 trials (9 blocks of 20 trials). The gain-, loss- and mixed blocks were pseudo-randomized such that the same block type was never presented twice in a row. The trials within a task block were also pseudo-randomized such that the same vase configuration was never presented more than 4 trials in a row. The participants had the opportunity to take a short break after each block. The experiment lasted ~ 50 minutes in total and participants received a base payment of 8 euros for their

participation. Participants were told that all the money they gained and lost would be added to or subtracted from this value. In the end, the task was set up in a way that participants would always receive a bonus of 50 cents on top of this base payment.

**Experiment 1B and 1C.** The procedure of Experiment 1B and 1C was similar to the procedure of Experiment 1A. Because our primary hypotheses concerned comparing gain events with loss events, we did not include any mixed trials in the experiment. The participants were presented with only two vase configurations: (1) 75%-25% vases: 15 marbles had one color and 5 marbles the other color and (2) 50%-50% vases: 10 marbles had one color and 10 marbles the other color. This was done to reduce differences in visual processing between the different vase configurations. Both colored marbles were associated with a monetary value that participants could gain or lose. In gain trials, the monetary values varied between 10 cents and 90 cents (in steps of 10 cents) and in loss trials the monetary values varied between -90 and -10 cents (in steps of 10 cents). In both trial types, all combinations of monetary values associated with red and blue marbles were possible, except for combinations of the same monetary values.

The first screen with the vase, the marbles and the monetary values associated with the marbles was presented for 4 s. Next, the same response screen as in Experiment 1A, during which participants could indicate their curiosity, was presented for 2.5 s. The response screen was followed by a blank screen with a jittered duration between 2.5 and 4.5 s (uniformly distributed) and an outcome screen (2 s). As in Experiment 1A, participants had a 50% chance that they would see the outcome (and their curiosity would be satisfied), and a 50% chance that the outcome was withheld. When the outcome was presented, the outcome screen depicted the selected marble in the middle of the screen and the amount of money participants won or lost in that trial below the marble. When the outcome was not presented, the screen depicted a black marble and question marks at the location of the monetary value. As such, the visual input was roughly comparable between trials in which the outcome was presented versus not presented. After a trial ended, there was a blank screen with a jittered duration between 3.5 and 4.5 s (uniformly distributed) and after every 9 trials, the duration of the blank screen was prolonged with an additional 16 s. As in Experiment 1A, participants had no way of influencing whether they would observe the outcome of a particular trial, or what the outcome of that trial would be. Again, participants were told that a marble would be selected on every trial and that they would earn or lose the money associated with that marble, even if the outcome was not presented. All money that they won or lost would be summed and added to or subtracted from the money they received for participation.

In Experiment 1B and 1C, the gain- and loss trials were pseudo-randomized such that the same trial type was never presented more than five times in a row. The participants completed 4 blocks of 54 trials (216 trials in total: 180 gain trials and 180 loss trials). Each vase configuration was presented on 108 occasions, so 54 times for each trial type (gain/loss). After each block, the participants had the opportunity to take a short break if they wanted. The experiment itself lasted ~ 70 minutes in total. Participants of Experiment 1B received a base payment of 12 euros and participants of the fMRI Experiment 1C received a base payment of 20 euros for their participation. Participants were told that all the money they gained and lost would be added to or subtracted from this value. In the end, the task was set up in a way that participants of both experiments would always receive a bonus of 50 cents on top of this base payment.

**Experiment 2.** In Experiment 2, we aimed to investigate participants' curiosity more implicitly by means of testing their willingness to wait to see the outcome [16]. We used this willingness to wait measure because it is well established to index an item's motivational value [25], which has previously been linked to curiosity [16, 20, 26]. Time is costly, so this willingness to wait measure allowed us to assess whether people would be willing to pay for non-

instrumental information even if it was costly in terms of their time. If participants decided to wait, they would always see the outcome whereas they would never see the outcome if they decided not to wait. By means of this measure, we can assess which types of information people want to approach and which types of information people want to avoid.

The willingness to wait task (Fig 1B) was in essence the same as the curiosity task of Experiment 1B and 1C. The first screen, depicting the vase, the marbles and the monetary values, was presented for 4 s. On the next screen, instead of giving a curiosity response, participants had to indicate whether they wanted to see the outcome of that trial ("Do you want to see the outcome?") by pressing either "Yes" or "No". Participants could give their responses on a button box, using their right index finger and right middle finger to respond "Yes" or "No" respectively. The response screen was presented until the participant responded, with a maximum of 4 s. If they pressed "No", a blank screen was presented briefly (0.5 s), followed by a screen on which the outcome was not presented (2 s). If they pressed "Yes", they saw a blank screen with a jittered duration between 3.5 and 6.5 s (uniformly distributed) before they saw a screen on which the outcome was presented (2 s). The participants had no influence on how long they had to wait for the outcome exactly, but they were aware that the waiting time varied between 3.5 and 6.5 seconds. When the outcome was presented, the outcome screen depicted the randomly selected marble in the middle of the screen and the amount of money participants won or lost in that trial below the marble. When the outcome was not presented, the screen depicted a black marble and question marks at the location of the monetary value. The next trial started after a jittered inter-trial interval between 1 and 2 s (uniformly distributed).

It should be noted that in this experiment, seeing the outcome was contingent on participants' decisions, such that they would always see the outcome when they were willing to wait, and never see the outcome when they were not willing to wait. As such, this measure gave us the opportunity to investigate for what type of lotteries participants decided to see the outcome and what type of information they wanted to avoid. However, participants were aware that a marble would be randomly selected on every trial and that they would always gain or lose the money associated with that marble, also when they did not want to wait to see the outcome. As in Experiment 1A, 1B and 1C, the money they won or lost on every trial would be summed and this sum of money would be added to or subtracted from the money they earned for their participation. This sum of money was depicted on the screen after the experiment had ended.

The gain and loss trials were pseudo-randomized such that the same trial type was never presented more than five times in a row. The participants completed 4 blocks of 54 trials (216 trials in total: 180 gain trials and 180 loss trials). Each vase configuration was presented on 108 occasions, so 54 times for each trial type (gain/loss). After each block, the participants could take a short break if they wanted. The total duration of Experiment 2 varied from ~60 to ~75 minutes and depended on the number of trials on which the participants chose to wait to see the outcome. Regardless of the task duration, participants received a base payment of 11.50 euros for their participation. They were told that all the money they gained and lost would be added to or subtracted from this value. In the end, the task was set up in a way that participants would always receive a bonus of 50 cents on top of this base payment.

## Experimental design & primary statistical analyses

Our primary hypotheses (see preregistration) concerned comparing positive events (gain trials) with negative events (loss trials). Therefore, the primary statistical analyses concerned the gain and loss trials only. We investigated whether there was a relationship between the main effects of outcome valence (gain/loss), outcome uncertainty, expected value and the curiosity ratings (Experiment 1A, 1B and 1C) or willingness to wait decisions (Experiment 2). In

addition, we investigated whether the effects of outcome uncertainty and expected value on curiosity/willingness to wait differed between the gain and loss trials by assessing the significance of interaction effects between outcome uncertainty and outcome valence, and expected value and outcome valence, on curiosity or willingness to wait. We did so using a combination of mixed effects modeling (reported here) and more classical repeated measures ANOVAs (reported in S1 Text). This allowed us to verify the robustness of our results and demonstrate that our conclusions do not depend on the analytical framework employed.

In order to do so, outcome uncertainty and expected value were calculated for every trial (X) as follows:

$$\text{Outcome Uncertainty}(X) = \sum\nolimits_{i=1}^{2} (x_i - \text{Expected Value}(X))^2 p_i \qquad (1)$$

$$\text{Expected Value}(X) = \sum\nolimits_{i=1}^{2} x_i p_i \qquad (2)$$

Here, $x_i$ is the monetary value associated with marble color (i) and $p_i$ the probability that this marble will be drawn. Hereby, outcome uncertainty (Eq 1) reflects the variance of the possible outcomes in trial (X) and expected value (Eq 2) reflects the mean expected reward contained in trial (X).

It should be noted that we used a different calculation for outcome uncertainty in one of our previous studies [16], and we initially preregistered to use that calculation of outcome uncertainty in the current manuscript as well. However, both metrics are almost identical, and variance is a more common measure of uncertainty (see for example [28, 29]). Therefore we decided to quantify outcome uncertainty as variance in the current manuscript.

Additionally, it should be noted that expected value is always positive in a gain trial and always negative in a loss trial. To be able to compare the effects of expected value between the gain and loss trials, we converted expected value to absolute expected value (such that—90 cents in a loss trial would be treated the same as + 90 cents in a gain trial etc.). However, the metric of interest here is the effect of reward magnitude (signed expected value) on curiosity, which is reflected in the interaction between outcome valence (gain/loss) and absolute expected value.

**Analyses using the BRMS package in R.**   Whereas we initially preregistered to perform the main statistical analyses of Experiment 1A using the clmm function of the ordinal package [30], we performed the statistical analyses for all experiments using the brm function of the BRMS package instead [31]. The BRMS package allowed us to fit Bayesian generalized (non-)linear multivariate multilevel models for full Bayesian inference. The main reason for performing the main analyses using the BRMS package, is that the BRMS package is robust and also suitable for assessing binomial dependent variables, such as the willingness to wait decisions (Experiment 2). Thus, the BRMS package allowed us to analyze the data from all reported experiments in a similar way.

The models of Experiment 1A, 1B and 1C included "curiosity rating (1–4)" as ordinal dependent variable, whereas the model of Experiment 2 included "willingness to wait (yes/no)" as binomial dependent variable. Models for both experiments included main effects of "outcome valence (gain/loss)", "outcome uncertainty" and "expected value (absolute)", and interaction effects between "outcome valence (gain/loss)" and "outcome uncertainty" and between "outcome valence (gain/loss)" and "expected value (absolute)" as fixed effects. The models included a full random effects structure [32, 33], so that a random intercept and random slopes for all effects were included per participant. The predictors for outcome uncertainty and expected value (absolute) were mean centered and scaled. We used the default

priors of the brms package (Cauchy priors and LKJ priors for correlation parameters; see [31]). The models were fit using four chains with 10000 iterations each (5000 warm up) and inspected for convergence. Coefficients were deemed statistically significant if the associated 95% posterior credible intervals were non-overlapping with zero.

If the interaction effects between "outcome valence (gain/loss)" and "outcome uncertainty" or between "outcome valence (gain/loss)" and "expected value (absolute)" were significant, we ran models on gain and loss trials separately. These models included "curiosity rating (1–4)" as an ordinal dependent variable, "outcome uncertainty" and "expected value (absolute)" as fixed factors, and a full random effects structure [32, 33]. In this way we could assess significance of outcome uncertainty and expected value (absolute) on curiosity of the gain and loss trials separately.

In addition to the analyses reported here, we performed similar analyses using repeated measures ANOVAs in SPSS (RRID:SCR_002865) and Bayesian repeated measures ANOVAs in JASP (RRID:SCR_015823). These analyses were performed to accommodate readers that are used to interpreting frequentist statistics instead of Bayesian credible intervals and to verify that our conclusions do not depend on the analytical framework employed (S1 Text).

### Data visualization

For Experiment 1A, we divided the levels of outcome uncertainty in six percentile bins per gain/loss condition, such that the 1st bin represents 1/6th of the lowest levels of outcome uncertainty, the 2nd bin represents the 1/6th– 2/6th of the lowest levels, etc. Similarly, absolute expected values were divided in six percentile bins (the 1st bin represents 1/6th of the lowest values, the 2nd bin the 1/6th– 2/6th of the lowest values, etc.). We did the same for Experiment 1B, 1C and for Experiment 2, except that the outcome uncertainty levels and the absolute expected values were divided in eight percentile bins per gain/loss condition. For each bin, we calculated a mean curiosity rating or mean percentage willingness to wait for gains and losses separately. Next, these mean scores per bin were averaged across participants and the standard error of the mean (SEM) was calculated for each bin of outcome uncertainty and absolute expected value of the gain and loss trials separately. Least squares lines illustrate the effects (Fig 2).

To illustrate to what extent individual participants showed sensitivity to gain versus loss trials, we calculated the mean curiosity ratings (Experiment 1A, 1B and 1C) and mean percentages willingness to wait (Experiment 2) for gain and loss trials separately per participant (Fig 3—left panels). Additionally, to illustrate to what extent participants were sensitive to the effect of outcome uncertainty, we calculated the difference in curiosity/willingness to wait scores between high and low outcome uncertainty for gain and loss trials separately per participant (Fig 3—middle panels). The same was done to illustrate the sensitivity to the absolute expected value effects by calculating the difference in curiosity/willingness to wait scores between high and low absolute expected value for gain and loss trials separately per participant (Fig 3—right panels).

### Results

### Experiment 1A, 1B and 1C

In line with the results of a previous study [16], Experiment 1A, 1B and 1C (Fig 2A–2C) showed that curiosity strongly increased with outcome uncertainty (**Exp. 1A** BRMS: 95% CI [.58,1.11]; **Exp. 1B** BRMS: 95% CI [1.17,2.60]; **Exp. 1C** BRMS: 95% CI [1.56,2.94]). There was also a main effect of outcome valence, such that curiosity was higher for gains (**Exp. 1A** $M$ = 2.69; $SD$ = .44; **Exp. 1B** $M$ = 2.99; $SD$ = 0.51; **Exp. 1C** $M$ = 2.82; $SD$ = .38) compared with

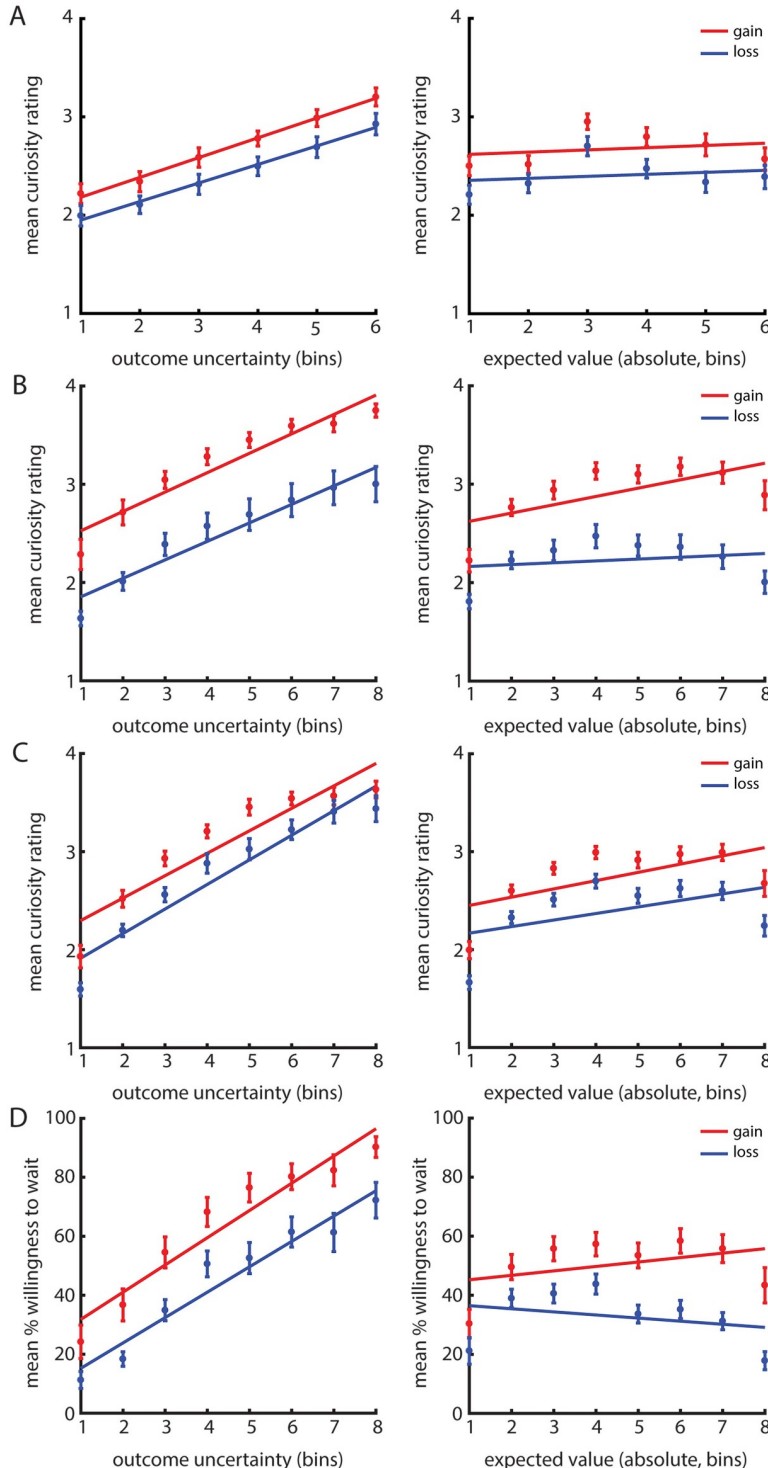

**Fig 2. Behavioural results of Experiment 1A (A) Experiment 1B (B), Experiment 1C (C) and Experiment 2 (D).** In all panels, the x-axis depicts percentile bins of the levels of outcome uncertainty (left) and the absolute expected values (right) for both gains (in red) and losses (in blue). The y-axis depicts the mean curiosity ratings for each percentile of outcome uncertainty and absolute expected value in the gain and loss context (A, B and C) or the % of trials in which participants were willing to wait for the outcome in the gain and loss context (D). The error bars depict the standard error of the mean (SEM) for each outcome uncertainty and absolute expected value bin of the gains or loss trials. For other details of the data visualization, see Methods – *2.5 Data Visualization*. **A**. Experiment 1A showed that curiosity

was higher for gains than for losses. Curiosity monotonically increased with increasing outcome uncertainty, which was not modulated by gain/loss outcome valence. There was no significant modulation of curiosity by (absolute) expected value for gains and losses. **B**. Experiment 1B showed that curiosity was higher for gains than for losses. Curiosity monotonically increased with increasing outcome uncertainty, which was not strongly modulated by gain/loss outcome valence. There was a significant modulation of curiosity by expected value for gain trials, but not for loss trials. **C**. Experiment 1C showed that curiosity was higher for gains than for losses. There was a monotonic increase of curiosity with increasing outcome uncertainty, which was not modulated by gain/loss outcome valence. There was a significant modulation of curiosity by absolute expected value for gain and for loss trials, such that participants were more curious about higher gains and losses. **D**. Experiment 2 showed that willingness to wait was higher for gains than for losses. Willingness to wait increased monotonically with outcome uncertainty, and there was no evidence for a modulation of this effect by gain/loss outcome valence. People were more willing to wait for higher compared with lower gains, but less willing to wait for higher compared with lower losses.

losses (**Exp. 1A** *M* = 2.43; *SD* = .48; BRMS: 95% CI [.18,.51]; **Exp. 1B** *M* = 2.29; *SD* = 0.58; BRMS: 95% CI [.61,1.86]; **Exp. 1C** *M* = 2.49; *SD* = .37; BRMS: 95% CI [.25,.71]).

Crucially, there was no interaction between outcome uncertainty and outcome valence In Experiment 1A and 1C (**Exp. 1A** BRMS: 95% CI [-.05,.09] **Exp. 1C** BRMS: 95% CI [-.05,.16]). The absence of an interaction was supported by the results from the repeated measures ANO-VAs (S1 Text) and the Bayes Factors of.21 (Exp. 1A) and.20 (Exp 1C) indicated that there is moderate evidence that an interaction between outcome valence and outcome uncertainty should not be included in the model. In Experiment 1B, there was evidence for an interaction between outcome uncertainty and valence (**Exp. 1B** BRMS: 95% CI [.08,.37]), but the effect appears to be less robust and present in a smaller proportion of participants (Fig 3B—middle panel). Moreover, the presence of this interaction was not supported by the results of the repeated measures ANOVA (S1 Text) and the Bayes Factor of.21 indicates that there is moderate evidence that the interaction term should not be included in the model.

Scatterplots show that the effect of outcome valence (Fig 3A–3C, left panels) as well as the effect of outcome uncertainty (Fig 3A–3C, middle panels) were highly robust and consistent across participants for all experiments. This indicates that whereas positive valence leads to higher curiosity, people are overall more curious when information gaps are larger. This is the case when these information gaps signal good news (gains) as well as when they signal bad news (losses).

There was no significant interaction between absolute expected value and outcome valence in Experiment 1A (**Exp. 1A** BRMS: 95% CI [-.09,.12]). Also, curiosity did not increase with absolute expected value in Experiment 1A (**Exp. 1A** BRMS: 95% CI [-.02,.26]), indicating that curiosity did not scale with reward magnitude. In Experiment 1B and 1C, however, there was a significant main effect of absolute expected value, such that curiosity was higher for high compared with low absolute expected value (**Exp. 1B** BRMS: 95% CI [.22,.47]; **Exp. 1C** BRMS: 95% CI [.26.55]). In Experiment 1B, there was also evidence for an interaction between absolute expected value and outcome valence (**Exp. 1B** BRMS: 95% CI [.15,.51]), indicating that the effects of absolute expected value differed between gain and loss trials. Indeed, analyses of the gain trials and loss trials separately, revealed a positive relationship between absolute expected value and curiosity in the gain trials, such that participants were more curious for higher compared with lower gains (**Exp. 1B** BRMS: 95% CI [.44,.94]). However, there was no relationship between absolute expected value and curiosity in the loss trials (**Exp. 1B** BRMS: 95% CI [-.17,.21]), such that there was no difference in curiosity about higher losses compared with lower losses. In Experiment 1C, there was no interaction between outcome valence and absolute expected value (**Exp. 1C** BRMS: 95% CI [-.02,.15]), indicating that participants were more curious about higher compared with lower gains and losses.

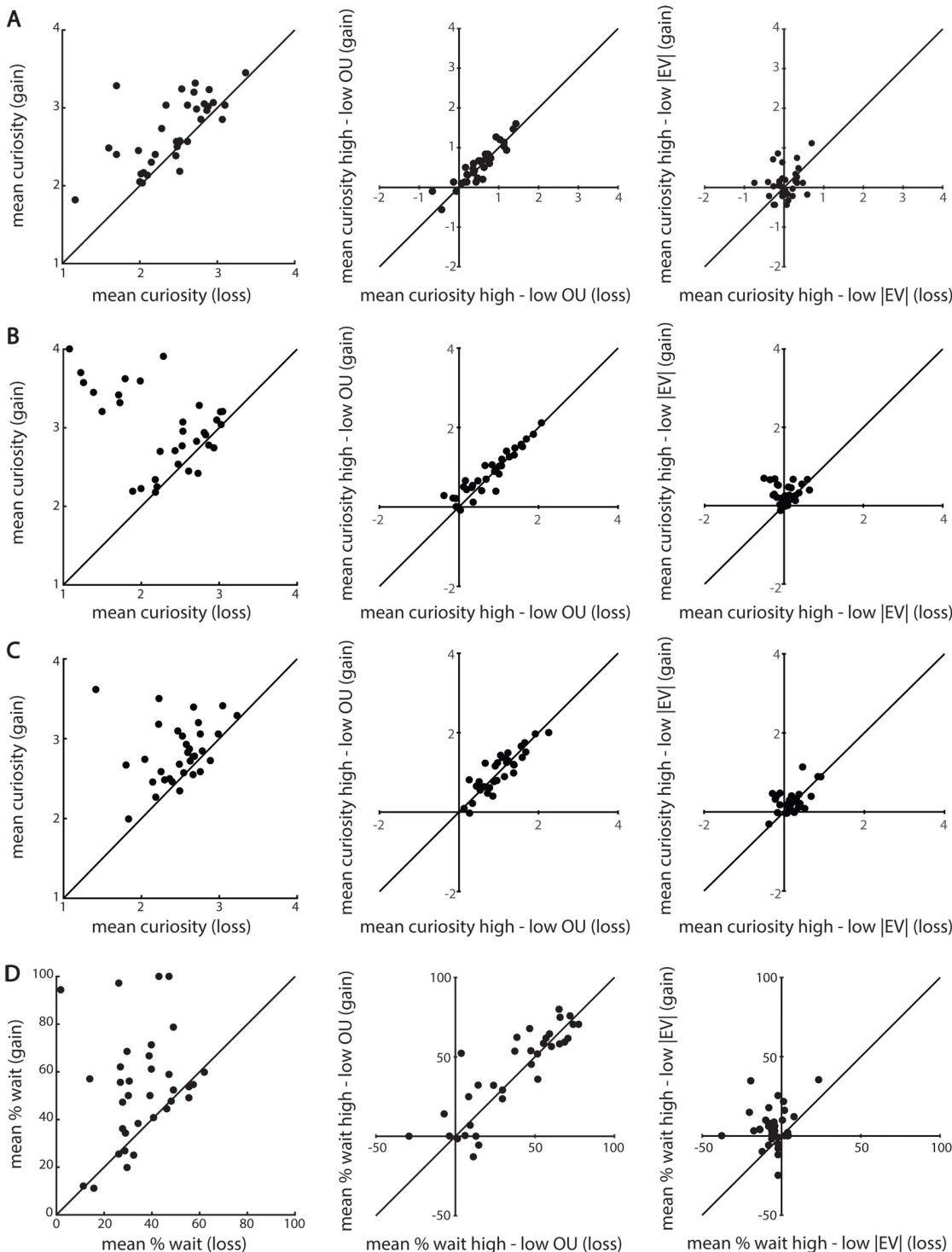

**Fig 3. Individual variability of the outcome valence (gain/loss), outcome uncertainty (OU) and absolute expected value (|EV|) effects of Experiment 1A (A,B,C) and Experiment 2 (D,E,F).** The left panels depict individual variability in participants' curiosity about gains compared with losses. In all panels, the x-axis depicts the mean curiosity/willingness to wait for the loss trials and the y-axis depicts the mean curiosity/willingness to wait for gain trails. Every dot depicts one participant. In all panels, most dots are above the diagonal, indicating that a great majority of the participants were more curious (**A, B** and **C**—left panels) and more willing to wait (**D**—left panel) for gains compared with losses. The middle panels depict individual variability in the extent to which participants were driven by outcome uncertainty for gain and loss trials. In all panels, the x-axis depicts the mean curiosity/ willingness to wait for high minus low outcome uncertainty in loss trials, with positive and negative values indicating a positive or

negative relationship between outcome uncertainty and curiosity / willingness to wait in loss trials respectively. The y-axis depicts the mean curiosity / willingness to wait for high minus low outcome uncertainty in gain trials, with positive and negative values indicating a positive or negative relationship between outcome uncertainty and curiosity / willingness to wait in gain trials respectively. Every dot depicts one participant. In all panels, the majority of participants have a positive relationship between outcome uncertainty and curiosity/willingness to wait for both gain and loss trials. The right panels depict individual variability in the extent to which participants were driven by absolute expected value for gain and loss trials. In all panels, the x-axis depicts the mean curiosity/willingness to wait for high minus low absolute expected value in loss trials, with positive and negative values indicating a positive or negative relationship between absolute expected value and curiosity / willingness to wait in loss trials respectively. The y-axis depicts the mean curiosity/willingness to wait for high minus low absolute expected value in gain trials, with positive and negative values indicating a positive or negative relationship between absolute expected value and curiosity/willingness to wait in gain trials respectively. Every dot depicts one participant. For Experiment 1A (**A**—right panel), most participants are clustered around 0, explaining the lack of an effect of absolute expected value in both gain and loss trials. In Experiment 1B (**B**—right panel), most participants have a positive relationship between absolute expected value and curiosity in the gain trials (as indicated by the positive y-values), whereas the relationship between absolute expected value and curiosity in the loss trials is more variable (as indicated by both negative and positive x-values). This explains the finding that participants were more curious about higher values of absolute expected value in gain trials and the lack of an absolute expected value effect in loss trials. In Experiment 1C (**C**–right panel), however, most participants have a positive relationship between absolute expected value and curiosity in gain trials (as indicated by positive y-values) as well as in loss trials (as indicated by positive x-values). This explains the finding that participants were more curious about higher absolute expected values in gain and loss trials. For Experiment 2 (**D**—right panel), most participants have a positive relationship between absolute expected value and willingness to wait in the gain trials (as indicated by the positive y-values). However, a majority of these participants have a negative relationship between absolute expected value and willingness to wait in the loss trials (as indicated by the negative x-values). This explains the finding that participants are more willing to wait for higher gains, but less willing to wait for higher losses.

## Experiment 2

In Experiment 2 (Fig 2D), we tested people's curiosity more implicitly by means of assessing their willingness to wait to see the outcome. On average, participants were willing to wait in 45.4% of all trials ($SD$ = 15.1). We found that willingness to wait increased with outcome uncertainty (**BRMS**: 95% CI [1.23,2.33]) and that the percentage willingness to wait was higher for gain ($M$ = 55.1; $SD$ = 23.6) compared with loss trials ($M$ = 35.6; $SD$ = 14.5; **BRMS**: 95% CI [.25,.98]). This shows that people not only indicated to be more curious (Experiment 1A, 1B and 1C), but that they were also more willing to wait to see the outcome (Experiment 2) and thus time costs of non-instrumental information seeking were offset when outcome uncertainty was higher and when anticipating gains versus losses. Both the effect of outcome valence (Fig 3D—left panel) as well as the effect of outcome uncertainty (Fig 3D—middle panel) were robust and present in a large proportion of participants. We found no evidence for an interaction between outcome uncertainty and outcome valence on willingness to wait (**BRMS**: 95% CI [-.01,.43]). The absence of an interaction was supported by the results from the repeated measures ANOVAs (S1 Text) and the Bayes Factor of .24 indicated that there is moderate evidence that an interaction between outcome valence and outcome uncertainty should not be included in the model.

There was a significant interaction between absolute expected value and outcome valence (**BRMS**: 95% CI [.08,.27]), but no main effect of absolute expected value on willingness to wait (**BRMS**: 95% CI [-.06,.12]). This indicates that the effects of absolute expected value differed between gain and loss trials. Indeed, analyses of gain and loss trials separately revealed a positive relationship between absolute expected value and willingness to wait in the gain trials (**BRMS**: 95% CI [.06,.37]), such that participants were more willing to wait for higher gains than for lower gains. However, there was a negative relationship between absolute expected value and willingness to wait in the loss trials (**BRMS**: 95% CI [-.25,-.02]), such that participants were *less* willing to wait for higher losses compared with lower losses (see also Fig 3D—right panel). In other words: participants were less willing to wait when they would lose more money. This finding replicates the earlier found relationship between expected value and participants' desire to know [12].

## Discussion

Humans are more curious when the information gap between what they know and what they don't know is larger [16]. Here, we examined whether this positive relationship between information gap and curiosity depends on whether the outcome will contain good or bad news. While it has been shown that people are curious about both losses and gains, the tendency to seek information is smaller for losses than gains, presumably because such knowledge boosts the anticipation of negative outcomes [12, 21, 22, 34, 35]. However, people are still curious about negative information, albeit to a lesser degree than about positive information (12,20). Also in the current studies, we found that curiosity increased with the uncertainty of information for both gains and losses. In addition to this effect, we found that people were overall more curious about gains compared with losses. However, the effect of outcome uncertainty on curiosity was not reliably different for gains and losses, indicating that these two effects seemed to operate largely independent of each other. This was the case for explicit curiosity-ratings as well as for more implicit willingness-to-wait decisions, demonstrating that participants are willing to seek non-instrumental information even if it is costly as a function of outcome uncertainty and valence. As such, these results highlight two separate motives underlying curiosity, reflecting both a drive to reduce uncertainty [16] as well as, separately, a drive to maximize positive versus negative information [12].

### Curiosity is related to uncertainty reduction

Participants were more curious about information that could reduce their uncertainty, in line with our previous study with a similar paradigm [16] as well as other work [13, 14, 18, 19, 36]. Moreover, Experiment 2 revealed that participants were even willing to pay with time to see the outcome of more uncertain lotteries, demonstrating that participants were willing to pay the price of time for non-instrumental information. The current work strengthens recent findings from Kobayashi and colleagues (2019), Charpentier and colleagues (2018), and our recent work [16] by showing that such effects of outcome uncertainty generalize to the domain of losses: Participants were not only curious about uncertain gains, but also about more uncertain losses. This demonstrates that participants show a general drive to reduce uncertainty, regardless of whether the information they could obtain contained good news (gains) or bad news (losses).

### Positive versus negative belief updating

In addition, we found in all experiments that participants were overall more curious about the outcome of gain trials compared with loss trials. This is in line with recent evidence indicating that participants are overall more curious about gains compared with losses [12] or more generally about positive compared with negative information [20]. Importantly, in these studies there was no absence of curiosity about losses: people were curious and willing to wait for obtaining information about losses, albeit to a lesser degree than for gains. Here we demonstrate that even in a passive observation paradigm, people still exhibit a preference for positive versus negative belief updating. Moreover, our findings extend that prior work [12, 20] by showing that the effect of valence is not modulated by the uncertainty of the outcome.

Our findings concur with a recent proposal [37] stating that people take "affect" into account when deciding what they want to know. This means that, all else being equal, people will choose to seek information when the affective response towards knowing is more positive than the affective response to remaining ignorant. In contrast to our findings using non-instrumental paradigms, studies using other paradigms in which rewards could be maximized by means of exploration (instrumental paradigms) showed that people exhibit higher

exploration rates for losses compared with gains [10, 11]. These findings can be explained by the notion of loss attention. That is, people show intensified alertness in the face of potential losses. Thus, exploration in instrumental contexts is greater for losses than gains, presumably because it allows participants to maximize the positive outcome of loss avoidance. By contrast, curiosity in non-instrumental contexts is greater for gains than losses, perhaps because it allows participants to maximize the positive affective state associated with knowing they have won. Therefore, we should emphasize that our conclusions do not generalize to situations where people seek information to maximize reward.

## Multiple drives represent independent mechanisms of curiosity

The current experiments did not provide convincing support for the hypothesis that the effect of outcome uncertainty was stronger for gains than losses (for replications see [38, 39]). This suggests that the bias to seek information about positive compared with negative events operates largely independently from the outcome uncertainty effect. Our results might thereby suggest that the information gain from resolving uncertainty is intrinsically valuable [13, 14, 20], regardless of whether the information is positive or negative. Another possible explanation is that it reflects an active motivation to resolve the aversive state of not knowing ([38, 40–42] see also [43]). Future research should shed light on which of these motives is dominant in people's desire for information and whether this differs between positive and negative contexts. Kobayashi and colleagues (2019) described in a recent paper that participants' behavior can be described as a mixture of motives related to uncertainty reduction and anticipatory utility. Also, Charpentier and colleagues (2018) found a boost in information seeking when the outcome was most uncertain, both for gains as well as for losses. Our findings extend this work by showing that an interaction between uncertainty and valence was not robust, suggesting that such multiple drives represent independent rather than interactive mechanisms of curiosity.

Whereas the effects of outcome uncertainty and gain/loss outcome valence on curiosity were robust and consistent over experiments, the effects of expected value on curiosity were more variable. In Experiment 1A, we found no effect of expected value on curiosity, such that participants were indifferent about the height of the gains and losses. This lack of an effect of expected value might reflect a failure to calculate expected it value in Experiment 1A at all (see also [16]). In Experiment 1B and 1C we found that participant were more curious about the outcome of lotteries with higher absolute expected value in gain trials (Experiment 1B) or in gain and loss trials (Experiment 1C). In Experiment 2, however, we found that participants were more willing to wait when they would gain more money, but *less* willing to wait when they would lose more money. This finding concurs with recent work by Kobayashi and colleagues (2019) and Charpentier and colleagues (2018), who found that participants choose knowledge about future desirable outcomes (when gains were higher) and choose to remain ignorant about future undesirable outcomes (when losses were higher). Participants in the latter study were even willing to pay for knowledge about gains and for ignorance about losses. In other words: we find that in the loss domain, willingness to wait increases with the size of the information gap, but decreases with the magnitude of the expected loss. Whereas we are driven by uncertainty reduction, there is an additional preference for anticipating positive outcomes (savouring), while avoiding the dread that might be associated with anticipating negative outcomes [12, 15, 44, 45].

## Limitations and future directions

In the current study, we used a non-instrumental lottery observation task, which enabled us to manipulate outcome valence, as well as outcome uncertainty and expected value in a

quantitative and controlled fashion. Prior work on human curiosity has made use of a trivia questions paradigm [20, 26, 46], in which participants have to rate their curiosity about trivia questions. These findings suggested that individuals are mostly curious about information of intermediate uncertainty [26], which is in contrast with our observation of a linear relationship between uncertainty and curiosity. However, it might reflect that participants in previous work were not curious about questions with highest uncertainty (i.e., lowest confidence) because they were simply not interested in the topics of these questions, confounding the relationship between curiosity and uncertainty. Whereas the tasks used in the current study might seem further removed from everyday situations that elicit curiosity, one of the benefits is that it did allow us to manipulate outcome valence and outcome uncertainty in a quantitative and controlled fashion. However, future studies could shed more light on the ecological validity of the current paradigm. This could for example be done by means of online studies in which performance on this task is combined with experience sampling of real-life fluctuating states of curiosity.

## Conclusion

Altogether, our findings advance the understanding of the psychological mechanisms underlying curiosity, by suggesting that there might be separate mechanisms that affect curiosity, related to uncertainty reduction (knowing) and processing reward context (savouring).

## Supporting information

**S1 Text. Analyses of Experiment 1A, 1B, 1C and 2 using repeated measures ANOVAs.**
(DOCX)

**S2 Text. Including the mixed outcome valence condition.**
(DOCX)

**S3 Text. Effects of uncertainty: Entropy or absolute difference?.**
(DOCX)

**S1 Table. Overview of Experiments 1A, 1B, 1C and 2.**
(DOCX)

**S1 Fig. Behavioural results of Experiment 1A, including the mixed condition.**
(DOCX)

## Author Contributions

**Conceptualization:** Lieke L. F. van Lieshout, Floris P. de Lange, Roshan Cools.

**Data curation:** Lieke L. F. van Lieshout.

**Formal analysis:** Lieke L. F. van Lieshout, Iris J. Traast.

**Funding acquisition:** Floris P. de Lange, Roshan Cools.

**Investigation:** Lieke L. F. van Lieshout, Iris J. Traast.

**Methodology:** Lieke L. F. van Lieshout, Floris P. de Lange, Roshan Cools.

**Project administration:** Lieke L. F. van Lieshout.

**Supervision:** Floris P. de Lange, Roshan Cools.

**Visualization:** Lieke L. F. van Lieshout.

**Writing – original draft:** Lieke L. F. van Lieshout.

**Writing – review & editing:** Iris J. Traast, Floris P. de Lange, Roshan Cools.

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
