## [Decision Letter · Decision Letter 0]

22 Feb 2021

PONE-D-20-34597

Curiosity or savouring? Information seeking is modulated by both uncertainty and valence

PLOS ONE

Dear Dr. van Lieshout,

Thank you for submitting your manuscript to PLOS ONE. After careful consideration, we feel that it has merit but does not fully meet PLOS ONE’s publication criteria as it currently stands. Therefore, we invite you to submit a revised version of the manuscript that addresses the points raised during the review process.

Reviewer 1 has some comments about placing the findings more properly in existing literature, which is important to address.

Reviewer 2 asks to include Experiment 1B and 1C in main text; I'll leave this decision to you, but please be explicit in your cover letter why you chose to do so or not. This reviewer also has some questions for clarification. 

I also had some minor comments, mainly for clarification; numbers refer to line numbers.

280: "the outcomes were contingent.." only what they see is contingent, not the actual outcome, right?

313: I suggest to already mention variance here (instead of “spread”)

336: Change from clmm to BRMS: These arguments about BRMS already applied before the preregistration. What then made the authors change specifically? Was the result from clmm less satisfying? I don’t mind changing the analysis method per se, but if you’re going to be so explicit about it, then please motivate the change.

380: Bayes factor: Please specify which model is in numerator and which in denominator.

We look forward to receiving your revised manuscript.

Kind regards,

Tom Verguts

Academic Editor

PLOS ONE

Journal Requirements:

2.Thank you for submitting the above manuscript to PLOS ONE. During our internal evaluation of the manuscript, we found significant text overlap between your submission and the following previously published works, some of which you are an author.

- https://www.biorxiv.org/content/10.1101/2020.10.13.337477v1

- https://www.sciencedirect.com/science/article/pii/S2352154620301248?via%3Dihub

- https://www.biorxiv.org/content/10.1101/195461v1.full

- https://www.predictivebrainlab.com/files/Lieshout_et_al_Jneuro.pdf

Please revise the manuscript to rephrase the duplicated text, cite your sources, and provide details as to how the current manuscript advances on previous work. Please note that further consideration is dependent on the submission of a manuscript that addresses these concerns about the overlap in text with published work.

Reviewers' comments:

Reviewer's Responses to Questions

**Comments to the Author**

1. Is the manuscript technically sound, and do the data support the conclusions?

Reviewer #1: Yes

Reviewer #2: Yes

2. Has the statistical analysis been performed appropriately and rigorously? 

Reviewer #1: Yes

Reviewer #2: Yes

3. Have the authors made all data underlying the findings in their manuscript fully available?

Reviewer #1: Yes

Reviewer #2: Yes

4. Is the manuscript presented in an intelligible fashion and written in standard English?

Reviewer #1: Yes

Reviewer #2: Yes

5. Review Comments to the Author

Reviewer #1: In this study “Curiosity or savouring? Information seeking is modulated by both uncertainty and valence” the authors show -as the title highlights – that curiosity is driven by both valence and uncertainty. They conclude that “Curiosity thus follows from multiple drives, including a drive to reduce uncertainty, as well as, separately, a drive to maximize positive information.” Curiosity is measured via rating and via willingness to wait. The authors task is an adaptation of a well validated task published several times before by this group. The analysis is straight forward and the data supports the conclusions. The paper is well written and a nice addition to the growing literature on curiosity.

My main concern is that the authors present the results as surprising and novel, while the same findings have been published a few times before. This does not mean that the current conceptual replication should not be published in this journal, but the authors should correct for this misrepresentation. There are also inaccurate statements regarding past studies and their relationship to the current one (see details below).

Previous studies have shown that curiosity is influenced by both uncertainty and valence, independently, in the same task. These papers include Kobayashi et al 2019; Kobayashi & Hsu 2019, Charpentier et al 2018.

For example, Charpentier et al 2018 employs a similar conceptual design assessing both explicit and implicit curiosity and shows that EV and uncertainty influence information seeking and curiosity both in the gain and loss domain, with greater information-seeking and curiosity in the gain domain. I quote the results on choice, but similar results are reported for explicit rating: “We formally tested this by running a general linear mixed-effect model predicting choices from EV and uncertainty (defined as the SD of the outcome distribution). This revealed the strong positive effect of EV on knowledge choice [estimate = 0.706 ± 0.196 (SE), t(4,176) = 3.60, P = 0.0003], as well as a smaller but significant effect of uncertainty [estimate = 0.196 ± 0.067 (SE), t(4,176) = 2.93, P = 0.0034] ... In other words, participants were more likely to seek knowledge when the likelihood of winning was high and losing was low, and there was a boost in information seeking when outcome probability was most uncertain (i.e., close to 0.5).”

The graph of the results of reference 12 looks very similar to the graph in this paper. This is not a problem for the current paper, it just has to be acknowledged.

The authors’ following statement is incorrect: “It has been reported that humans tend to avoid negative information, presumably because such knowledge boosts the anticipation of negative outcomes (12,21,22,34,35). Contrary to this notion, we found that curiosity increased with the uncertainty of information for both gains and losses. “ (line 490). If you look at the figure from reference 12 you can see that even in the loss domain the likelihood of selecting knowledge over ignorance is above 50%. That is, even when participants expect bad news they are still curious, it is just that they are less curious than when they expect good news. This is true for real life information seeking such as medical screenings as well. Most people do want to know the results of their tests even if they expect bad news, but they are more ambivalent about this. Moreover, information-seeking (as quoted above) did alter with both uncertainty and EV in the loss domain in reference 12. This was shown in the analysis and it is why there is a curvature in the figure above.

The corrections to the manuscript should be relatively straight forward.

Reviewer #2: This manuscript by van Lieshout et al. examines how curiosity, or a desire to seek non-instrumental information for its own sake, is shaped by possible future outcomes. The authors conducted multiple experiments in which human participants rated their curiosity about, or decided whether to wait for, the outcome of monetary lotteries. They showed that, across experiments, participants are more curious about the outcome when the lottery generates a gain rather than a loss, and when the outcome variance is larger.

In my opinion, this manuscript makes a significant contribution to active research on information seeking by providing clear empirical evidence on factors that shape curiosity. The experiments were nicely designed and executed, the data was throughly analyzed and reported, and the manuscript is generally well written and very easy to understand. I also appreciate that the authors tested an alternative account that curiosity is driven by information-theoretic uncertainty (entropy) by model comparison (S4 text). Therefore, I would recommend this manuscript for publication.

Below are my suggestions on how to make this manuscript even stronger upon publication, all of which are minor:

1. I see no reason not to include Experiments 1B and 1C in the main text (currently in Supplementary Texts and Figures). They are equally powered as the main Experiment 1, and if anything, Experiment 1B is more similarly designed to Experiment 2. By showing the results of Experiments 1B and 1C alongside Experiment 1, the authors can demonstrate the robustness of the main findings. More importantly, it would also help readers interpret the reported effect of lottery expected value appropriately (looking at the results across experiments, it seems to me that the effect of EV might exist, even if it is very weak). At least, I hope that the authors explicitly spell out the findings on the EV effect from 1B and 1C in the main text.

2. Experiments 1 and 2 have a few important differences in task design. The authors emphasize that, in Experiment 2, participants’ choices actually determined whether the information was delivered or not. Another important difference is that, unlike Experiment 1, information seeking was costly; by choosing to seek information, participants had to spend more time on the experiment. While it is not a monetary cost, effort cost, or opportunity cost, I still think it is an important empirical evidence that participants are willing to seek costly yet non-instrumental information (depending on outcome valence and variance), and I would recommend that the authors emphasize it a bit more.

3. Was there any overlap between participants across experiments?

4. P. 7, lines 146-148: I am confused by this sentence. Did the authors originally aim at n=34 and collected a few more in case they needed to exclude some? Or are the authors simply claiming that the resultant n is enough?

5. P. 21, line 536: “In these studies” - which studies?

6. S3 Text, line 1: “Furthermore” can be omitted.

7. S4 Text: In leave-one-out cross validation, what was left out? A trial? A participant?

6. PLOS authors have the option to publish the peer review history of their article (what does this mean?). If published, this will include your full peer review and any attached files.

Reviewer #1: No

Reviewer #2: No

---

## [Author Response · Author response to Decision Letter 0]

20 Jul 2021

I have pasted the responses from the document "response to the reviewers" below:

Editor’s comments:

Thank you for submitting your manuscript to PLOS ONE. After careful consideration, we feel that it has merit but does not fully meet PLOS ONE’s publication criteria as it currently stands. Therefore, we invite you to submit a revised version of the manuscript that addresses the points raised during the review process.

Thank you for giving us the opportunity to revise the paper. Based on the two insightful reviews and the thoughtful comments of the editor, we have revised the paper. We gave detailed explanations on every point in the current document and we adjusted the manuscript accordingly.

1. 280: "the outcomes were contingent.." only what they see is contingent, not the actual outcome, right?

As the editor correctly states, whether participants would see the outcome or not (Experiment 2) was contingent on participants’ willingness to wait decisions, and not the actual outcome. To avoid confusion, we have changed the sentence on page 14 to: 

“It should be noted that in this experiment, seeing the outcome was contingent on participants’ decisions, such that they would always see the outcome when they were willing to wait, and never see the outcome when they were not willing to wait.”

2. 313: I suggest to already mention variance here (instead of “spread”)

We agree with the editor’s suggestion and we have changed this sentence on page 16 as follows:

“Hereby, outcome uncertainty (Equation 1) reflects the variance of the possible outcomes in trial (X) and expected value (Equation 2) reflects the mean expected reward contained in trial (X)”

3. 336: Change from clmm to BRMS: These arguments about BRMS already applied before the preregistration. What then made the authors change specifically? Was the result from clmm less satisfying? I don’t mind changing the analysis method per se, but if you’re going to be so explicit about it, then please motivate the change.

The main argument to change from clmm to BRMS is that the BRMS package (in contrast to clmm) is also suitable for assessing binomial dependent variables (i.e. the willingness to wait decisions in Experiment 2). It should be noted that we only preregistered Experiment 1A and its statistical analyses and not the follow-up willingness to wait experiment (Experiment 2). To be consistent in our analysis method for all experiments, we decided to analyze the data for both experiments with the same package. Using BRMS is also consistent with the data analysis for one of our other studies (van Lieshout, de Lange & Cools, 2020, PsyArXiv), for which we did preregister to analyze the data with the BRMS package in R. 

We now motivate this change in more detail on page 16 - 17 as follows: 

“Whereas we initially preregistered the main statistical analyses of Experiment 1A using the clmm function of the ordinal package (30), we performed the statistical analyses for all experiments using the brm function of the BRMS package instead (31). The BRMS package allowed us to fit Bayesian generalized (non-)linear multivariate multilevel models for full Bayesian inference. The main reason for performing the main analyses using the BRMS package, is that the BRMS package is robust and also suitable for assessing binomial dependent variables, such as the willingness to wait decisions (Experiment 2). Thus, the BRMS package allowed us to analyze the data from all reported experiments in a similar way.”

4. 380: Bayes factor: Please specify which model is in numerator and which in denominator.

The Bayes Factors that are mentioned in the manuscript are obtained using the method as described in Supplement 1 under “Bayesian Repeated Measures ANOVAs in JASP”. In short: we use model averaging across matched models to get a single Bayes Factor for each effect in the repeated measures ANOVA. As such, each Bayes Factor reflects the change from prior to posterior inclusion odds, and can intuitively be understood as the amount of evidence that the data gives for including the corresponding experimental factor in a model. 

This model averaging across matched models was first described in this blogpost by Sebastian Mathôt: https://www.cogsci.nl/blog/interpreting-bayesian-repeated-measures-in-jasp and later implemented by the developers of JASP. This method first identifies all the models that contain the effect of interest, but no interactions with the effect of interest (the “with” models). Next, the method identifies all the models that are “stripped” of the effect of interest (the “without” models). The resulting Bayes Factor is calculated as the sum of P(model|data) of all “with” models (numerator), divided by the sum of p(model|data) of all “without” models (denominator). 

We now explain this in more detail in Supplement 1 as follows:

“In JASP (RRID:SCR_015823), we performed the Bayesian equivalent of the repeated measures ANOVAs reported above. We used the default Cauchy prior to compute Bayes Factors for each effect. For interpretability of analyses with multiple factors, we used model averaging across matched models to get a single BF for each effect in the repeated measures ANOVA. Specifically, each BF is computed as the sum of P(model|data) of all models containing the effect of interest (but no interactions with the effect of interest), divided by the sum of P(model|data) of all the models that are stripped of this effect of interest. As such, each Bayes Factor reflects the change from prior to posterior inclusion odds and can intuitively be understood as the amount of evidence that the data gives for including an experimental factor in a model (BFincl). The BFincl will converge to zero when the factor should not be included in the model, or to infinity when the factor should be included in the model. Values close to one indicate that there is not enough evidence for either conclusion.”

When we talk about Bayes Factors in the main text of the manuscript, we now clearly refer to this explanation in Supplement 1. 

 

Reviewer #1: 

In this study “Curiosity or savouring? Information seeking is modulated by both uncertainty and valence” the authors show -as the title highlights – that curiosity is driven by both valence and uncertainty. They conclude that “Curiosity thus follows from multiple drives, including a drive to reduce uncertainty, as well as, separately, a drive to maximize positive information.” Curiosity is measured via rating and via willingness to wait. The authors task is an adaptation of a well validated task published several times before by this group. The analysis is straight forward and the data supports the conclusions. The paper is well written and a nice addition to the growing literature on curiosity.

We thank the reviewer for the overall positive evaluation of the manuscript. 

My main concern is that the authors present the results as surprising and novel, while the same findings have been published a few times before. This does not mean that the current conceptual replication should not be published in this journal, but the authors should correct for this misrepresentation. There are also inaccurate statements regarding past studies and their relationship to the current one (see details below).

Previous studies have shown that curiosity is influenced by both uncertainty and valence, independently, in the same task. These papers include Kobayashi et al 2019; Kobayashi & Hsu 2019, Charpentier et al 2018. For example, Charpentier et al 2018 employs a similar conceptual design assessing both explicit and implicit curiosity and shows that EV and uncertainty influence information seeking and curiosity both in the gain and loss domain, with greater information-seeking and curiosity in the gain domain. I quote the results on choice, but similar results are reported for explicit rating: “We formally tested this by running a general linear mixed-effect model predicting choices from EV and uncertainty (defined as the SD of the outcome distribution). This revealed the strong positive effect of EV on knowledge choice [estimate = 0.706 ± 0.196 (SE), t(4,176) = 3.60, P = 0.0003], as well as a smaller but significant effect of uncertainty [estimate = 0.196 ± 0.067 (SE), t(4,176) = 2.93, P = 0.0034] ... In other words, participants were more likely to seek knowledge when the likelihood of winning was high and losing was low, and there was a boost in information seeking when outcome probability was most uncertain (i.e., close to 0.5).” 

We thank the reviewer for pointing this out. Previous studies have indeed assessed the effects of both uncertainty and valence on curiosity (e.g. Charpentier et al., 2018; Kobayashi et al., 2019; Kobayashi & Hsu, 2019). For example, Kobayashi and colleagues (2019) have demonstrated that a subset of participants was sensitive to uncertainty in both the gain domain as well as the loss domain. However, unlike our study, the authors do not directly test whether the effect of outcome uncertainty differs between the gain and the loss domain (i.e. they don’t assess possible interactions between uncertainty and valence). They do state, however, that the parameters of uncertainty were highly correlated for gains and losses, indicating that participants followed similar sampling strategies for gains and losses. 

Charpentier and colleagues (2018) demonstrate a boost in information seeking when the outcome of a lottery is uncertain. This is the case for gain as well as for loss lotteries. However, again, unlike our paper, this paper also does not directly examine the interaction between uncertainty and valence (gains versus losses). 

Nevertheless, we do agree with the reviewer that, in the original manuscript, we overstated the novelty of our study in the sense that previous studies have looked at the effects of uncertainty and valence on curiosity in the same study. We have now adjusted the text in the manuscript to better highlight that (i) our work builds on this previous work, and (ii) the main aim of our study was to directly test the interaction between outcome uncertainty and valence on page 6 as follows: 

“Previous studies have addressed the existence of multiple motives for curiosity. Different individuals exhibit various mixtures of effects of uncertainty and expected value, both for gains as well as for losses (15). Additionally, previous work has demonstrated a boost in information seeking when outcome probability was most uncertain (12), both for gains as well as for losses. However, these studies have not explicitly addressed the interaction between uncertainty and valence on non-instrumental curiosity. It is this interaction that allows us to investigate the degree to which effects of uncertainty and valence represent independent or interactive mechanisms of non-instrumental curiosity.”

And on page 25 as follows:

“Kobayashi and colleagues (2019) described in a recent paper that participants’ behavior can be described as a mixture of motives related to uncertainty reduction and anticipatory utility. Also, Charpentier and colleagues (2018) found that a boost in information seeking when the outcome was most uncertain, both for gains as well as for losses. Our findings extend this work by showing that an interaction between uncertainty and valence was not robust, suggesting that such multiple drives represent independent rather than interactive mechanisms of curiosity.”

The graph of the results of reference 12 looks very similar to the graph in this paper. This is not a problem for the current paper, it just has to be acknowledged.

We agree with the reviewer that the graph in this paper (Figure 2) looks similar to the results of Charpentier and colleagues (2018). We have now acknowledged this in the results section on page 23 of the manuscript as follows.

“This finding replicates the earlier found relationship between expected value and participants’ desire to know (12).”

The authors’ following statement is incorrect: “It has been reported that humans tend to avoid negative information, presumably because such knowledge boosts the anticipation of negative outcomes (12,21,22,34,35). Contrary to this notion, we found that curiosity increased with the uncertainty of information for both gains and losses. “ (line 490). If you look at the figure from reference 12 you can see that even in the loss domain the likelihood of selecting knowledge over ignorance is above 50%. That is, even when participants expect bad news they are still curious, it is just that they are less curious than when they expect good news. This is true for real life information seeking such as medical screenings as well. Most people do want to know the results of their tests even if they expect bad news, but they are more ambivalent about this. Moreover, information-seeking (as quoted above) did alter with both uncertainty and EV in the loss domain in reference 12. This was shown in the analysis and it is why there is a curvature in the figure above.

We agree with the reviewer that the statement should be formulated more carefully. Also in our data, we find that participants are still curious for losses, albeit to a lesser degree than for gains. We have now reformulated the sentence quoted by the reviewer (page 23) above to:

“While it has been shown that people are curious about both losses and gains, the tendency to seek information is smaller for losses than gains, presumably because such knowledge boosts the anticipation of negative outcomes (12,21,22,34,35). However, people are still curious about negative information, albeit to a lesser degree than about positive information (12,20). Also in the current studies, we found that curiosity increased with the uncertainty of information for both gains and losses.”

The corrections to the manuscript should be relatively straight forward.

Reviewer #2: 

This manuscript by van Lieshout et al. examines how curiosity, or a desire to seek non-instrumental information for its own sake, is shaped by possible future outcomes. The authors conducted multiple experiments in which human participants rated their curiosity about, or decided whether to wait for, the outcome of monetary lotteries. They showed that, across experiments, participants are more curious about the outcome when the lottery generates a gain rather than a loss, and when the outcome variance is larger.

In my opinion, this manuscript makes a significant contribution to active research on information seeking by providing clear empirical evidence on factors that shape curiosity. The experiments were nicely designed and executed, the data was throughly analyzed and reported, and the manuscript is generally well written and very easy to understand. I also appreciate that the authors tested an alternative account that curiosity is driven by information-theoretic uncertainty (entropy) by model comparison (S4 text). Therefore, I would recommend this manuscript for publication.

Below are my suggestions on how to make this manuscript even stronger upon publication, all of which are minor.

We thank the reviewer for the positive evaluation of our manuscript. We respond to the comments of the reviewer below. 

1. I see no reason not to include Experiments 1B and 1C in the main text (currently in Supplementary Texts and Figures). They are equally powered as the main Experiment 1, and if anything, Experiment 1B is more similarly designed to Experiment 2. By showing the results of Experiments 1B and 1C alongside Experiment 1, the authors can demonstrate the robustness of the main findings. More importantly, it would also help readers interpret the reported effect of lottery expected value appropriately (looking at the results across experiments, it seems to me that the effect of EV might exist, even if it is very weak). At least, I hope that the authors explicitly spell out the findings on the EV effect from 1B and 1C in the main text.

We agree with the reviewer that it will be clearer to include the results of Experiment 1B and Experiment 1C in the main text as well. We have now adjusted this in the manuscript. 

2. Experiments 1 and 2 have a few important differences in task design. The authors emphasize that, in Experiment 2, participants’ choices actually determined whether the information was delivered or not. Another important difference is that, unlike Experiment 1, information seeking was costly; by choosing to seek information, participants had to spend more time on the experiment. While it is not a monetary cost, effort cost, or opportunity cost, I still think it is an important empirical evidence that participants are willing to seek costly yet non-instrumental information (depending on outcome valence and variance), and I would recommend that the authors emphasize it a bit more.

We agree with the reviewer that the empirical finding that participants were willing to seek costly, non-instrumental information should be emphasized more throughout the manuscript. 

We have now emphasized this finding more in the methods section, when introducing the experiment on page 13, as follows:

“In Experiment 2, we aimed to investigate participants’ curiosity more implicitly by means of testing their willingness to wait to see the outcome (16). We used this willingness to wait measure because it is well established to index an item’s motivational value (25), which has previously been linked to curiosity (16,20,26). Time is costly, so this willingness to wait measure allowed us to assess whether people would be willing to pay for non-instrumental information even if it was costly in terms of their time.” 

We have also emphasized the finding in the results section on page 22 as follows:

“This shows that people not only indicated to be more curious (Experiment 1A, B and C), but that they were also more willing to wait to see the outcome (Experiment 2) and thus time costs of non-instrumental information seeking were offset when outcome uncertainty was higher and when anticipating gains versus losses.”

As well as in the discussion on page 23 and on page 24 as follows:

“However, the effect of outcome uncertainty on curiosity was not reliably different for gains and losses, indicating that these two effects seemed to operate largely independent of each other. This was the case for explicit curiosity-ratings as well as for more implicit willingness-to-wait decisions, demonstrating that participants are willing to seek non-instrumental information even if it is costly as a function of outcome uncertainty and valence. As such, these results highlight two separate motives underlying curiosity, reflecting both a drive to reduce uncertainty (16) as well as, separately, a drive to maximize positive versus negative information (12).”

“Moreover, Experiment 2 revealed that participants were even willing to pay with time to see the outcome of more uncertain lotteries, demonstrating that participants were willing to pay the price of time for non-instrumental information.”

3. Was there any overlap between participants across experiments?

There was no overlap between participants across experiments. Participants who participated in one of the studies described in the manuscript were excluded from participation in any of the other studies. We have emphasized this on page 9 in the participants section of the methods:

“It should be noted that there was no overlap between participants across experiments. Participants who participated in one of the studies, were excluded from participation in any of the other experiments reported in the manuscript.”

4. P. 7, lines 146-148: I am confused by this sentence. Did the authors originally aim at n=34 and

collected a few more in case they needed to exclude some? Or are the authors simply claiming that the resultant n is enough?

We understand the confusion of the reviewer regarding the following sentence: “The sample size of N = 34 was chosen to detect a within-subject effect of at least medium size (d>0.5) with 80% power using a two-tailed one-sample or paired t-test.”

We were indeed aiming for N=34 included data sets, as also preregistered. To make this clearer we have adjusted this sentence as follows (see page 8):

“We were aiming for a sample size of N = 34 included datasets to detect a within-subject effect of at least medium size (d>0.5) with 80% power using a two-tailed one-sample or paired t-test.”

5. P. 21, line 536: “In these studies” - which studies?

“These studies” are referring to the studies conducted in the current manuscript. We have adjusted the sentence as follows: 

“The current experiments did not provide convincing support for the hypothesis that the effect of outcome uncertainty was stronger for gains than losses (for replications see (38,39)). This suggests that the bias to seek information about positive compared with negative events operates largely independently from the outcome uncertainty effect. Our results might thereby suggest that the information gain from resolving uncertainty is intrinsically valuable (13,14,20), regardless of whether the information is positive or negative.”

6. S3 Text, line 1: “Furthermore” can be omitted.

We agree with the reviewer and have now omitted “Furthermore” from the sentence in Supplement 2 (which was Supplement 3 in the previous version) as follows:

“We performed additional analyses on the data of Experiment 1A that were not of primary interest, but relevant to understand the data.”

7. S4 Text: In leave-one-out cross validation, what was left out? A trial? A participant?

The approximate leave-one-out cross validation (LOO), leaves out one data point (trial) each time.

We have now made this clearer by adjusting the text in Supplement 3 (which was Supplement 4 in the

previous version) as follows:

“Next, we performed model comparisons using the loo method of the loo package (4) for approximate leave-one-out cross validation (LOO) using (Pareto-smoothed) importance sampling (PSIS). The LOO Information Criterion (LOOIC) of the models are reported here, since they have the same purpose as the AIC, which are used for the model comparisons of Experiment 1. LOOIC is intended to estimate the expected log predictive density (ELPD) for a dataset. The ELPD is based on the sum of the logs of the leave-one-out predictive density given the data without one of the data points (one of the trials). The difference in ELPD between the models gives an indication of how well the models explain the data.”

---

## [Editor Report · Decision Letter 1]

23 Aug 2021

Curiosity or savouring? Information seeking is modulated by both uncertainty and valence

PONE-D-20-34597R1

Dear Dr. van Lieshout,

We’re pleased to inform you that your manuscript has been judged scientifically suitable for publication and will be formally accepted for publication once it meets all outstanding technical requirements.

Kind regards,

Tom Verguts

Academic Editor

PLOS ONE

Additional Editor Comments (optional):

Minor edit: “found that a boost” —> “found a boost”
---

## [Editor Report · Acceptance letter]

17 Sep 2021

PONE-D-20-34597R1 

Curiosity or savouring? Information seeking is modulated by both uncertainty and valence. 

Dear Dr. van Lieshout:

I'm pleased to inform you that your manuscript has been deemed suitable for publication in PLOS ONE. Congratulations! Your manuscript is now with our production department. 

Kind regards, 

on behalf of

Dr. Tom Verguts 

Academic Editor

PLOS ONE